# Effectiveness of More Personalized, Case-Managed, and Multicomponent Treatment for Patients with Severe Schizophrenia Compared to the Standard Treatment: A Ten-Year Follow-Up

**DOI:** 10.3390/jpm12071101

**Published:** 2022-07-04

**Authors:** Juan J. Fernández-Miranda, Silvia Díaz-Fernández, Francisco López-Muñoz

**Affiliations:** 1Asturian Mental Health Service Área V.-H.U. Cabueñes, Servicio de Salud del Pº de Asturias (SESPA), 33211 Gijón, Spain; marmotillazz@gmail.com; 2Asturian Institute on Health Research (ISPA), 33011 Oviedo, Spain; 3Faculty of Health Sciences, University Camilo José Cela, 28692 Madrid, Spain; flopez@ucjc.edu; 4Neuropsychopharmacology Unit, Hospital 12 de Octubre Research Institute, 28041 Madrid, Spain

**Keywords:** schizophrenia, treatment, personalized, outcomes, case management, hospital admissions, adherence, suicide, long-acting antipsychotic

## Abstract

Case management is a model of personalized intervention in people with severe mental illness. To explore the treatment adherence and effectiveness of patients with severe schizophrenia (Clinical Global Impression Severity, CGI-S ≥ 5) undergoing treatment in a community-based, case-managed program (CMP) with an integrated pharmacological and psychosocial approach compared with the standard treatment, an observational, ten-year follow-up study was conducted on patients treated in mental health units (MHUs) or a CMP (*n* = 688). Treatment discontinuation, hospitalizations, suicide attempts, and antipsychotic (AP) medications were recorded. Clinical severity was assessed with the CGI-S. Adherence to the CMP was higher than adherence to standard treatment (*p* < 0.001). There were fewer hospitalizations and suicide attempts in the CMP (*p* < 0.001). The clinical severity decreased more in the CMP (*p* < 0.005). Long-acting injectable (LAI) antipsychotic medication was more closely related to these outcomes than oral antipsychotics (APs) were (*p* < 0.001). Patients with severe schizophrenia in an integrated CMP recorded higher treatment compliance and better outcomes compared with standard care. Treatment with LAI APs was linked to these outcomes. A personalized combination of case management and LAI AP medication was more effective in these patients than standard treatment and oral APs.

## 1. Introduction

Case management (CM) is a model of community intervention for people with severe mental illness that provides a comprehensive range of therapeutic and rehabilitation interventions [1,2]. Intensive CM (ICM) highlights the need for small caseloads (fewer than 20) and a high-intensity approach. The core purposes of ICM are to improve outcomes, reduce hospital admissions, and uphold contact with services [1,2,3]. A comprehensive approach within ICM usually involves a range of psychosocial and pharmacological interventions [3,4]. CM includes an individualization of the therapeutic pathway for people with more severe schizophrenia, adapting to their needs at each moment and, therefore, adapting the intensity of the contact and the interventions provided [5,6].

Some studies have reported that ICM, when compared with standard care, reduces the loss of contact with health services, the severity of the symptoms, and hospital admissions while improving patients’ social performance [1,4,5,6]. However, the discussion continues about its effectiveness and superiority over other care models [1,7] since the evidence on its outcomes is not robust and it is difficult to reach overall conclusions [1,5,6,8,9]. An integrated approach in case management usually implies several psychosocial modalities and medications and is regarded as standard practice in treating patients with schizophrenia [1,2,3,4,5,6].

Relapses, measured by hospital (re)admissions, are a common criterion for the effectiveness of treatment for people with schizophrenia [10,11,12,13], as are suicide attempts (albeit to a lesser extent) [14]. Their relationship with the type of treatment (both in psychosocial and pharmacological aspects) has yet to be clearly established [1,10,14,15,16]. A high prevalence of non-adherence among patients with schizophrenia, especially those with greater clinical severity, seems to be associated with relapses and (re)hospitalizations [10,11,12,13,15,17] and with the risk of suicide [14,18,19]. Some studies suggest that variables such as program adherence may be associated with the effectiveness of CM [1,4,10,12].

Various factors may affect compliance to treatment in people with schizophrenia, including those related to general treatment (e.g., community-based or hospital-based) [3,4,10,11,12] and, specifically, AP medications (side-effects, kind of administration, etc.) [13,15,16,17,20,21,22]. Long-acting injectable antipsychotics (LAI APs) may therefore be considered an effective treatment strategy to improve compliance and reduce relapses and hospital admissions [16,21,22]. An area of interest is whether or not they can indirectly decrease suicide attempts [14,18,19].

There are few studies on these key aspects of the treatments for people with schizophrenia in real-world settings, especially for seriously ill patients; existing studies have used small samples and short follow-up times. Further research is still needed to confirm whether the ICM approach and LAI APs confer advantages over standard treatment and oral APs (OAPs) in terms of improved adherence and a reduced risk of admissions and suicide. This research sought to focus on these aspects in routine clinical practice. Specifically, the aim was to assess treatment adherence, clinical severity, hospital admissions, suicidal behavior, and the impact of the route of administration of Aps on the outcomes in a group of patients with schizophrenia treated in an intensive and integrated case management program (CMP) for patients with severe mental illness (SMI) compared with those receiving standard treatment in mental health units (MHUs).

## 2. Method

An observational, longitudinal study (ten-year follow-up) of people with severe schizophrenia (ICD 10: F-20; Clinical Global Impression-Severity, CGI-S ≥ 5) was carried out. The aim was to compare the effectiveness of an integrated case-managed approach (case-managed program, CMP) to the standard treatment administered in MHUs. The present study was carried out in Gijon (Spain) from January 2010 to December 2019.

The end of treatment was recorded, along with the AP medication prescribed and types of regimes (oral vs. LAI), psychiatric hospital admissions, and documented suicide attempts over ten years. The severity of illness was measured with CGI-S at the start and at 1, 2, 5, and 10 years of follow-up. In this research, non-adherence was considered as MHU or CMP treatment discontinuation for more than one month. In addition, hospitalization is defined as patient admissions associated with psychiatric symptoms.

### 2.1. Intervention

The studied CMP for patients with SMIis based on the principles of community care with intensive case management and long-term multicomponent intervention providing a wide range of treatment and rehabilitation services. The program aims to help people with SMI improve their psychosocial functioning, receive support for engaging in community life, develop coping skills, and ensure continuity of care.

Patients are referred to the CMP from their MHUs or general hospital psychiatric wards. The CMP provides comprehensive treatment for schizophrenia with 24-h assistance, day hospital, ambulatory care, and home care. The multidisciplinary treatment team is made up of psychiatrists, clinical psychologists, specialist mental health nurses, social workers, and occupational therapists. Specialist mental health nurses are usually the case managers (intensive: 20 patients per nurse). The program provides integrated psychological and pharmacological treatment, cognitive remediation, social skills training, self-care management training, psychoeducation, vocational intervention, and home support. In addition, it adapts the contact with different professionals and the intensity of the clinical and rehabilitative interventions to the patients’ needs and demands at all times. Patients are offered at least one individual consultation every 15 days with a clinician and at least weekly with their nurse/case manager. Group interventions (psychoeducation, cognitive remediation, and social skills) occur fortnightly and with a duration of 50 min for each session.

The CMP team is the primary source of care for its patients. It is situated away from the hospital. The CMP team meets three times a week. Part of the team is always available 24 h a day. There is daily (Monday to Friday, 8:00 to 15:00 h) availability of a psychiatrist, a psychologist, and a nurse on the team for any patient. All patients are assigned a psychiatrist, a nurse (in general, the case manager), and a psychologist. The CMP offers unlimited time for its services.

In the current study, intensive case management is defined as treatment in which patients receive care that is based on CM models with a caseload of up to 20 people and frequent contact with the patients. Standard care is when patients receive outpatient, non-intensive, and non-specific care. Standard care intervention is modeled on a generalist model provided by a community mental health service. There is no presence of specialized services, such as rehabilitation services.

The research was carried out according to the Declaration of Helsinki (WMA) and was approved by the Asturian Ethical Committee on Clinical Research (Spanish National Health Service) (P.I. 88/16).

### 2.2. Patients

The participants (*n* = 688; 344 in each group) were 18 years old and over and had a diagnosis of schizophrenia with serious symptomatology and impairment with a CGI-S score of ≥5. The treating clinicians decided whether the patients were likely to benefit from CMP treatment or whether they should remain in standard care in their MHUs. Participants were selected between January 2007 and December 2009. Half of the sample was treated in MHUs and the other half was in the CMP. The initial recruitment involved 700 subjects, but complete data were not available for some of them. All of them (or their legal guardians, as appropriate) signed informed consent before the treatment began.

The mean age of the participants was 43.4 (SD: 11.4) years; 62.2% were men and 37.8% were women. There were no significant differences between the two groups’ sociodemographic data and clinical characteristics, with only mild differences in age and the length of illness (higher in the CMP sample; *p* < 0.05). Table 1 shows the participants’ sociodemographic and psychiatric backgrounds. All the participants in the study had been treated before with at least one AP.

### 2.3. Data Analysis

A statistical analysis (descriptive and inferential) was carried out. Chi^2^ was used for qualitative variables (sex, type of AP), and the McNemar test was used to compare paired proportions (% of patients with suicide attempts, % of patients with hospital admissions, and % with LAI APs). For quantitative variables (number of suicide attempts, hospital admissions, and GCI-S scores), Student’s *t*-test for paired data was conducted. The confidence interval was set at 95%. The “R Development Core Team” program (version 3.6.1) MASS package (7.3-45 version) was used to process the data.

## 3. Results

The CGI-S mean score of MHU and CMP at the beginning of the study was 5.6 (standard deviation, SD: 0.7); in the MHUs, it was 5.3 (0.4); in the CMP, it was 5.9 (0.5) (*p*< 0.05). After 10 years, the CGI-S was 3.9 (1.1) in the MHU group and 3.1 (0.9) in the CMP group (*p* < 0.005) (Table 2). MHUs treated 61.1% of patients with SGAs (second-generation antipsychotics), while CMP treated 93% of patients with SGAs (*p* < 0.0001); 27.6% of patients in MHUs were on LAIs, compared to 57.6% of patients in the CMP (*p* < 0.001) (Table 2). Gender did not influence these prescriptions, as no significant differences were found.

A total of 43.6% of the patients discontinued treatment in MHUs, compared to 12.1% of patients in the CMP (*p* < 0.0001) (Table 2). Treatment discontinuation was closely linked to OAPs in both cases (*p* < 0.001) (Table 3). In the MHUs, 46.5% of the patients had at least one hospital admission, with an average of 3.2 (3.4) admissions; 9.9% were non-voluntary, with an average of 0.5 (0.3) admissions. In the CMP, 17.4% of the patients were admitted to hospitals, with an average of 0.9 (0.3) admission; 1.4% were non-voluntary, with an average of 0.01 (0.2) admissions. Both the differences in the number of admissions and in their voluntary nature between MHUs and the CMP were clearly statistically significant (*p* < 0.0001) (Table 2).

Regarding hospital admissions and the kind of AP therapy, being on first-generation antipsychotics (FGA) and not SGA treatment had an influence during the follow-up (*p* < 0.001), although the majority of patients received SGAs. Hospital admissions were linked to treatment with OAPs in MHUs (*p* < 0.001) and especially in the CMP (*p* < 0.0001). Taking OAPs made it more likely that admission would be involuntary (*p* < 0.0001) (Table 3).

Suicide attempts were significantly fewer in the CMP: 85 patients in MHUs and 20 patients in the CMP (*p* < 0.0001) (Table 2). There was no relationship between suicide attempts and whether the patient was taking FGA or SGA medication. However, there was a significant relationship between suicide attempts and oral AP treatment (vs. LAI), both in MHUs (*p* < 0.01) and particularly in the CMP (*p* < 0.0001) (Table 3).

Although CGI-S scores were higher in the CMP compared with the MHUs group at the beginning of treatment (*p* < 0.05), there was a significantly sharper decrease in scores in the CMP group at the end of the follow-up compared with those in MHUs (*p* < 0.005) (Table 2). In both groups, the drop in score was significant as of the first year of treatment, but it was higher in the CMP group (Figure 1).

Gender was not related to treatment adherence, hospital admissions or involuntariness, suicide attempts, the severity of illness, or the type of AP treatment (no significant relationships were found).

## 4. Discussion

### 4.1. Treatment Adherence

Treatment discontinuation of people with schizophrenia worsens the course of the disease and overall clinical severity [10,11,12,13,23], and it ultimately increases the risk of relapses and suicide [14,15,17,18,19,24,25]. Several studies suggest that compliance with treatment may be linked to the higher effectiveness of the CM approach, especially when the CMP is more intensive [6,7,9,12]. However, difficulty persists in assessing approaches that are broadly different despite sharing a common model [1,5,6]. The treatment compliance recorded in this research was higher than the compliance reported by most of the studies conducted on patients with schizophrenia, including those incase management programs [1,2,5,12,13]. The illness severity of the participants must be highlighted, as it often translates into treatment discontinuation [11,15,23,26].

In general, the use of SGA LAIs to improve compliance and treatment results in patients with schizophrenia is recommended [21,22,27,28,29,30]. Nevertheless, a consensus has not yet been established: meta-analyses comparing LAI APs with oral APs have provided conflicting results [31,32,33,34,35] because of randomized clinical trial (RCT) biases. RCTs involve patients with less severe symptoms, and frequent monitoring may also improve adherence [27,32,36]. Greater compliance with LAI AP and their increased effectiveness over oral APs is clearer in mirror and cohort studies [12,14,27,32,37], even at high doses [38,39,40].

Comparing this study’s findings with those of other studies on similar subjects and approaches [1,5,6,7] reveals that the reasons that CMP is significantly more effective than the MHU treatment (e.g., seven times fewer treatment discontinuations when LAIs were used only twice as often as OAPs) are possibly linked not only to closer and more intensive contact with patients but also to the integrated psychosocial and pharmacological treatment provided [3,4].

### 4.2. Hospital Admissions

One of the key purposes of CM is to reduce hospital admissions [6,7,13], and ICM has shown a reduction in the number of psychiatric admissions compared with standard care [1,5,6]. In turn, SGA LAIs have shown higher effectiveness to prevent relapse and hospitalization [22,26,27,28,41], which is better supported by naturalistic studies [16,26,27,28,32,34,35,42,43,44,45] than by a meta-analysis [34], particularly with regard to SGA LAIs [27,42,43,44,45]. Some naturalistic studies have reported fewer hospital admissions at high SGA LAI doses in patients with severe and resistant schizophrenia [38,39,40].

Hospital admissions are considered an indicator of relapse and severe clinical ecompensation [46,47]. In the current study, the number of hospital admissions arising in the CMP was significantly lower than the number of admissions for standard treatment and also that reported in previous studies with this patient profile [5,12,16,27,35,42,43,44,45]. It is also remarkable that there were fewer involuntary admissions in the CMP group than in the MHU group; this could suggest a better therapeutic relationship. Furthermore, the number of involuntary admissions was significantly lower among patients on LAI APs, mainly SGAs, than among those on OAPs. The decrease in CGI-S scores (clinical severity) in the CMP was significant and possibly linked to the decrease in hospitalizations. This points toward a relationship between the use of LAIs in ICM and the decrease in hospital admissions, especially involuntary ones.

### 4.3. Suicide Attempts

Compared with the general population, people with schizophrenia have an 8.5 times greater risk of suicide [25,48]. CM treatments with an interdisciplinary approach and CM seem to be effective models for preventing suicidal behavior [1,6,14], which is a priority in these programs; suicidal behavior is also regarded as an indicator of a program’s effectiveness [6,7,8]. CM programs may have an indirect impact on the prevention of suicide, as they help improve compliance and decrease relapses [1,5,6,7,14].

There is little evidence to suggest that antipsychotic medications lower the risk of suicide. The existing evidence appears to be the most favorable for SGAs, particularly clozapine and SGA LAIs [14,18,19,25]. Clozapine and SGA LAIs have the highest rates of treatment adherence and relapse prevention in people with schizophrenia [20,21,22,26], although their association with suicide prevention is yet to be determined [14,25]. Some meta-analyses did not record any significant differences between LAIs and oral APs with respect to all causes of death, or, specifically, to suicide [13,21,49]. By contrast, recent studies have revealed that LAIs are associated with a lower risk of death (including suicide) than oral APs [14,25,50].

The number of suicide attempts in the CMP was lower than that usually found in studies on patients with schizophrenia, not only those receiving standard treatments but also patients in specific programs for people with a severe illness [14,50]. The lower number of suicide attempts in the CMP could be the result of improved treatment adherence during the program [7,9,51].

### 4.4. Clinical Severity

CM seems to reduce the clinical severity of patients with schizophrenia [1,6,7]. When an integrated approach is adopted incase management using several psychological, pharmacological, and rehabilitation interventions, the severity of the illness is reduced [3,4,8]. In turn, the use of LAI APs seems to be an effective way to reduce the severity of symptoms [21,22,27,35,37].

The clinical severity, measured with the CGI-S, of patients receiving treatment in the CMP decreased significantly compared with the standard treatment in MHUs. The drop in illness severity in the CMP may be related to the lower number of hospital admissions and suicide attempts. Furthermore, at the end of the follow-up, the CGI-S scores of patients receiving LAI AP treatment were lower than for patients taking OAPs, especially in the CMP. Although the baseline CGI-S score was lower in the MHU group at the beginning of the study, this did not seem to be a clinically significant difference.

### 4.5. Effectiveness

Compared with standard care, ICM is effective in improving some treatment outcomes in people with SMI, although there is still a need to confirm its effectiveness over other models of continuity of care or over standard care [1,3,6,7,8]. Taking into account evidence with a low to moderate quality, ICM is sufficiently effective at improving outcomes in people with severe schizophrenia. In comparison with standard care, ICM may decrease hospitalizations and increase adherence and social functioning [1,2,5,6,7,9]. The program studied here includes an integrated treatment that comprises a higher intensity of psychosocial interventions than many other case-managed programs. This study suggests that this kind of approach clearly improves treatment adherence and decreases hospital admissions and suicide attempts. The reduction in CGI-S scores (clinical severity) was also significant and possibly linked to the decrease in hospital admissions and suicidal behavior.

LAI APs are a safe, tolerable [35,52] and effective treatment option; they prevent relapses and hospital admissions and are a strategy for improving adherence [24,31,32,36,37,53]). However, most recent studies comparing SGA OAPs with LAIs in severely ill patients are relatively scarce, and they have the disadvantage of a brief follow-up and relatively small samples [3,4,8,9,11,12]. In our study, treatment compliance was significantly higher in the CMP than in MHUs, although dropouts were significantly higher for both treatments inpatients on OAPs than in those treated with LAI APs. Our research also shows that LAI APs were associated with sharp reductions in hospital admissions and suicide attempts compared to OAPs.

### 4.6. Study Limitations and Strengths

The study’s design allowed us to compare standard care with care carried out in the CMP with a ‘naturalistic’ approach. Nevertheless, some limitations must be taken into account. This is an open-label, non-randomized design under clinical routine conditions. Strictly speaking, there is no control group, but a similar group of sociodemographic and clinical characteristics is used as an active comparator. We are aware that this design could present a bias in favor of the program studied. Moreover, to measure a clinically meaningful change in severity, we used the CGI-S, which is a non-specific instrument.

Although there are clearly many interventions within the comprehensive and integrated approach that the program follows with patients and that influence the program’s final results, AP medication was chosen here because of its importance in general treatment, its debatable effectiveness regarding the type of AP and the route of administration, and its most objective measurement. In fact, more patients were on LAIs in the ICM group than in the MHUs group, and this may be a confusing factor (i.e., did LAIs or CM improve the outcomes more?).

More severely ill patients may show greater improvement (regression to the mean), but this fact does not seem to have been a factor in the clinical severity differences between CMP and MHU patients. Finally, all the participants in this research were labeled as severely ill by the CGI-S. Our results may not be generalizable to mildly ill populations.

The main strength of this investigation is that it provides a snapshot of real-world results based on routine clinical practice, avoiding the abovementioned biases of RCTs. It is also the first study to assess the treatment adherence and effectiveness of a CMP vs. standard care in MHUs, as well as oral APs vs. long-acting injectables, in a wide group of severely ill patients in a real-world context over a long period.

## 5. Conclusions

The significant differences found in our study between standard treatment and a community-based program using ICM methodology, in terms of treatment adherence and clinical outcomes, allowed us to consider the CMP more effective than the standard treatment provided in MHUs. Our findings showed that specific strategies designed to increase adherence (e.g., community programs with ICM) decreased clinical severity, relapses, and suicidal behavior. Moreover, our findings showed that treatment with LAI APs clearly contributed to the achievement of these results.

The widespread implementation of community-based programs with intensive case management, integrated multicomponent treatment, and the use of long-acting antipsychotics should be considered as an effective treatment for people with severe schizophrenia who are at high risk of treatment discontinuation.

In short, this study shows that personalizing the treatment of patients with severe schizophrenia through case management achieves much greater adherence and better clinical results than standardized care from mental health units, which is less individualized. In addition, it highlights the importance of personalizing pharmacological treatment with drugs that favor adherence and clinical improvement in these people.

## Figures and Tables

**Figure 1 jpm-12-01101-f001:**
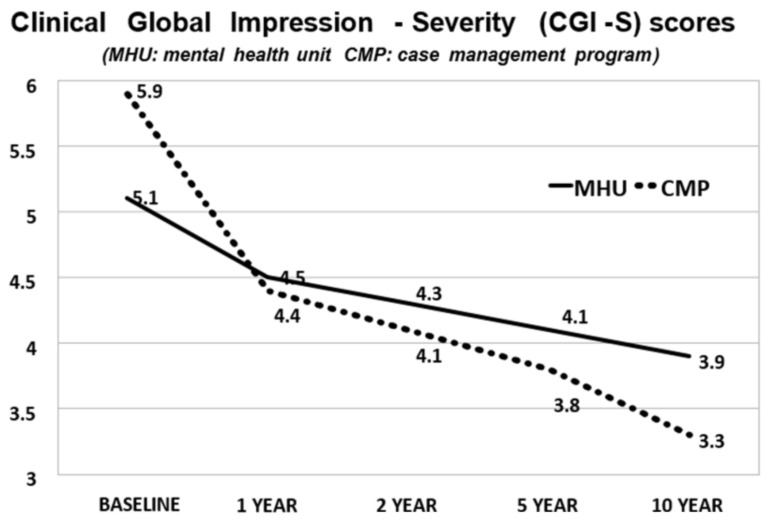
CGI-S Scores during the Follow-up. Mhu vs. Cmp.

**Table 1 jpm-12-01101-t001:** Patients’ Sociodemographic and Clinical Characteristics.

	Total (*n* = 688)	Mental Health Units (*n* = 344)	Severe Mental Illness Program (*n* = 344)
Gender: male (*n*, %)	427 (62.2)	210 (61.1)	217 (63.1)
Age (Av (SD)) years *	43.4 (11.4)	41.6 (10.9)	46. (11.9)
Length of illness (Av (SD)) years *	21.1 (5.8)	20.3 (4.9)	21.8 (5.2)
Previous tt duration (Av (SD)) years	11.2 (7.1)	10.1 (5.2)	13.3 (8.5)

(*n*, %): number, percentage; Av(SD): average (standard deviation); tt: treatment; *: *p* < 0.05.

**Table 2 jpm-12-01101-t002:** Clinical Treatment Outcomes after 10-Year Follow-Up.

*n* = 688	Mental Health Units (*n* = 344)	Case Management Program (*n* = 344)	χ^2^; *p*-Value
Treatment discontinuation (*n* (%))	150 (43.6)	42 (12.1)	26.16; <0.0001
CGI-S (Av (SD))	3.9 (1.1)	3.1 (0.9)	7.63; <0.005
FGA vs. SGA (%)	137(39.8)/210(61.1)	6 (1.7)/320 (93)	31.12; <0.0001
Oral/LAI AP (%)	249(72.4)/95(27.6)	149 (43.3)/198 (57.6)	8.91; <0.001
Hospitalization (*n* (%))	160 (46.5)	60 (17.4)	10.54; <0.0001
Hospitalization (Av (SD))	3.2 (3.4)	0.9 (0.3)	13.23; <0.0001
Involuntary hospital. (*n* (%))	34 (9.9)	5 (1.4)	28.01; <0.0001
Involuntary hospital. (Av (SD))	0.5 (0.3)	0.01 (0.2)	21.31; <0.0001
Suicide attempt (*n* (%))	85 (24.7)	20 (5.8)	10.54; <0.0001
*n* suicide attempts (Av (SD))	0.3 (0.1)	0.07 (0.02)	11.32; <0.0001

*n*: number of patients; %: percentage of patients; Av: average; SD: standard deviation; CGI-S: Clinical Global Impression-Severity scale FGA: First Generation Antipsychotics SGA: Second Generation Antipsychotics LAI. Long-acting injectable AP: antipsychotic.

**Table 3 jpm-12-01101-t003:** Type of Antipsychotic (Oral vs. Lai), Treatment Discontinuation, Hospital Admissions, and Suicide Attempts.

*n* = 688 (*n*(%))	Mental Health Units (*n* = 344)		Case Management Program (*n* = 344)	
Treatment discontinuation ***	150 (43.6)		30 (8.7)	
	*OAP*	*LAI-AP*	*OAP*	*LAI-AP*
Treatment discontinuation	120 (48.2)	30 (31.6) **	25 (16.8)	5 (2.5) ***
Hospital admissions **	160 (46.5)		60 (17.4)	
	*OAP*	*LAI-AP*	*OAP*	*LAI-AP*
Hospital admissions	130 (52.2)	30 (31.6) **	42 (28.4)	18 (9.2) ***
Involuntary admissions ***	34 (9.9)		5 (1.4)	
	*OAP*	*LAI-AP*	*OAP*	*LAI-AP*
Involuntary admissions	28 (11.2)	6 (6.3) **	4 (2.7)	1 (0.5) ***
Suicide attempts ***	85 (24.7)		20 (5.8)	
	*OAP*	*LAI-AP*	*OAP*	*LAI-AP*
Suicide attempts	67 (26.9)	18 (18.9) *	15 (10.1)	5 (2.5) ***

*: *p* < 0.01; **: *p* < 0.001; ***: *p* < 0.0001;(*n*, %): number, percentage; AP: antipsychotic; FG: first generation; SG: second generation; OAP: oral antipsychotic; LAI AP: long-acting injectable antipsychotic.

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
