# Peer review of "Effectiveness of More Personalized, Case-Managed, and Multicomponent Treatment for Patients with Severe Schizophrenia Compared to the Standard Treatment: A Ten-Year Follow-Up"

_jpm, 2022, doi:10.3390/jpm12071101_

Round 1

Reviewer 1 Report

In this longitudinal study of patients with schizophrenia, the authors compared two groups of patients receiving either standard treatment or intensive case-management (ICM) on a number of clinical outcomes.

The manuscript lacks some of the depth expected of papers describing psychosocial interventions. While the authors focused on describing assessment and evaluation of clinical outcomes, they glanced over their description of psychosocial treatments, especially IGM since that was a tailored intervention used in the study. The authors should spend more time on describing the actual ICM intervention (e.g. frequency, duration of sessions, duration of treatment etc.). In the Introduction, the authors mentioned assessing patients’ reasons for treatment discontinuation, yet I could not see any mention of it in the manuscript. If this paper is to inform real world clinical practice, it needs to include more details on the clinical methods that were used. A better description of standard treatment would be helpful too.

Additionally, do authors have any sense if some components of ICM intervention worked better than other components? Given extensive resources required for ICM it is unlikely that this intervention is feasible to implement in all settings. More commentary about this would be useful.

The manuscript could be improved by addressing some minor stylistic and structural issues (e.g. multiple paragraphs that describe related ideas should be merged; redundancies should be avoided), which I’ve attempted to outline in detail by including comments and edits in the manuscript document (please see attached. Briefly, the manuscript flow could be much improved by careful editing as both punctuation and grammar (e.g. the authors use a combination of present and past tense in a single section and often paragraph) need a little bit of work. I understand that it is not easy to produce a manuscript in the non-native language, which is why attempted to provide some structural and stylistic editing, hence so many comments in text.

Best of luck!

Author Response

  • The intervention description has been more detailed. We spent more time on describing the ICM intervention and, also, on a better description of standard treatment. We have included more details on the clinical methods that were used.
  • In the Introduction, and also in the Discussion, we have clarified what means treatment discontinuation
  • In Limitations we briefly discuss if some components of ICM intervention worked better than other components (mainly AP route of administration).
  • We have addressed (and sincerely we appreciate) all stylistic and structural issues, punctuation and grammar, outlined in detail by the reviewer. We have tried to improve the manuscript flow following all his/her comments and suggestions

All these changes are marked up using the “Track Changes” function, such that any changes can be easily viewed by the reviewer.

Reviewer 2 Report

The manuscript is a longitudinal observational cohort study comparing case-managed programs (CMP) with mental health units (MHUs). The author also focused on the role of antipsychotics (APs), considering oral versus long-acting injections (OAPs vs. LAIs) and first versus second-generation antipsychotics (FGA vs. SGA). In general, the study is relevant and provides insight into the topic of severely ill patients with schizophrenia. The availability of studies investigating relevant outcomes such as relapse, hospitalization, and suicidality in the very long term is scarce. The availability of methods used to combine observational and experimental designs in the context of evidence synthesis would make the results of the present valuable study in the future, regardless of the limitations of the study design. Nevertheless, the study could not be considered for publication at this stage. Here it follows my suggestions to improve the manuscript quality:

- Even if the overall meaning of the manuscript is understandable, the hypotactic sentencing compromises the reading flow. Therefore, the manuscript needs an anglophone proofreader revision.

- The aggregated results comparing CMP and MHUs suffer from a confounding bias of OAPs vs. LAIs and FGAs vs. LAI. More advanced statistical analyses should consider the impact of those factors on the long-term efficacy of the psychosocial intervention. Moreover, the authors report baseline differences that could be accounted for in the statistics and that are not. We suggest the author consider a bio-statistician revision of the results. It is a pity that such a large amount of data is not analyzed more articulately. Please, consider to report crude data in the appendix anyhow, it might be needed for data extraction of future evidence synthesis projects.

- The intervention description should be fully detailed. It means that the reader should be able to replicate at least the intervention. This implies that the authors should provide references the author can look up or an extended description of the sub-intervention adopted in the CMP in the manuscript appendix. 

- What does "deficient social conditions or support were considered" (line 113) mean? In which sense was it considered?

- in both the figures, the color legend is missing; please add it. And please, remove the yellow background in figure 2.

- the discussion might be shortened and revised according to new statistics derived from advanced statistics analyses (ANOVA and regressions).

Author Response

- We have tried to improve the reading flow. We have made changes in stylistic and structural issues, and also in punctuation and grammar.

- We have clarified results comparing OAPs vs. LAIs and FGAs vs. SGAs trying to avoid potential biases. Moreover, we discuss about it in Limitations.

- Baseline differences that could influence the results have been explained and discussed

- The intervention description has been more detailed. We spend more time on describing the ICM intervention and, also, on a better description of standard treatment. We have included more details on the clinical methods that were used. We have provided previous publications about it in References.

- "deficient social conditions or support were considered" (line 113) has been deleted, to avoid any confusion about its meaning.

- Both Figures have been erased: all information is in the Tables or in the text, and now they are redundant.

- The Discussion has been revised.

All these changes are marked up using the “Track Changes” function, such that any changes can be easily viewed by the reviewer.

Round 2

Reviewer 1 Report

Attached here is the revised manuscript with some further, minor edits and suggestions re. grammar and structure. 

Author Response

Attached you will find the new versión with all changes  suggested made.

Thank you
